# Increased Asian aerosols drive a slowdown of Atlantic Meridional Overturning Circulation

Fukai Liu [1,9] ✉, Xun Li [1,9], Yiyong Luo [1] ✉, Wenju Cai [1,2,3,4], Jian Lu [5], Xiao-Tong Zheng [1], Sarah M. Kang [6], Hai Wang [1] & Lei Zhou [7,8]

Observational evidence and climate model experiments suggest a slowdown of the Atlantic Meridional Overturning Circulation (AMOC) since the mid-1990s. Increased greenhouse gases and the declined anthropogenic aerosols (AAs) over North America and Europe are believed to contribute to the AMOC slowdown. Asian AAs continue to increase but the associated impact has been unclear. Using ensembles of climate simulations, here we show that the radiative cooling resulting from increased Asian AAs drives an AMOC reduction. The increased AAs over Asia generate circumglobal stationary Rossby waves in the northern midlatitudes, which shift the westerly jet stream southward and weaken the subpolar North Atlantic westerlies. Consequently, reduced transport of cold air from North America hinders water mass transformation in the Labrador Sea and thus contributes to the AMOC slowdown. The link between increased Asian AAs and an AMOC slowdown is supported by different models with different configurations. Thus, reducing emissions of Asian AAs will not only lower local air pollution, but also help stabilize the AMOC.

The Atlantic Meridional Overturning Circulation (AMOC) transports vast amounts of energy to the northern high latitudes, playing a pivotal role in modulating the global climate[1–3]. Increasing lines of evidence indicate that the AMOC is declining over the modern climate record[4–7], although debates persist[8–10]. Furthermore, numerical model simulations consistently project a slowdown or even collapse of the AMOC as a response to increasing greenhouse gases (GHGs) in the 21st century, as summarized by the Sixth Assessment Report of the United Nations Intergovernmental Panel on Climate Change (IPCC AR6)[11].

However, GHGs are not the only external factor influencing AMOC strength, and anthropogenic aerosols (AAs) have also been found to play an important role (Supplementary Fig. 1a)[12–18]. Compared to the globally well-mixed GHGs, AAs are highly variable in space and time due to their short residence time in the atmosphere. In contrast to the

monotonically increasing GHGs since the industrial revolution, there is a two-stage temporal evolution of the global mean AAs (Supplementary Fig. 1b): prior to the late 1980s, the global mean AAs shows a monotonic increase due to ramping aerosol emissions from both the eastern and western hemispheres; after the early 1990s it remains stable due to a reversal of the emission trend in Europe and North America (an effect of the Clean Air Act). By contrast, the AAs emission over South and East Asia continues to increase.

A growing number of literature recognizes the role of AAs in AMOC variations over the past century[12–18]. In particular, a multi-model of single-forcing simulations, known as the Detection and Attribution Model Intercomparison Project (DAMIP)[19], reveals a two-stage evolution of the AAs-induced AMOC change[15]: a monotonic steady strengthening before the mid-1990s and a rapid weakening

[1]Frontiers Science Center for Deep Ocean Multispheres and Earth System, Physical Oceanography Laboratory, and Sanya Oceanographic Institution, Ocean University of China, Qingdao, China. [2]Laoshan Laboratory, Qingdao, China. [3]State Key Laboratory of Marine Environmental Science & College of Ocean and Earth Sciences, Xiamen University, Xiamen, China. [4]State Key Laboratory of Loess and Quaternary Geology, Institute of Earth Environment, Chinese Academy of Sciences, Xi'an, China. [5]Atmosphere, Climate, and Earth Sciences Division, Pacific Northwest National Laboratory, Richland, USA. [6]Max Planck Institute for Meteorology, Hamburg, Germany. [7]School of Oceanography, Shanghai Jiao Tong University, Shanghai, China. [8]Southern Marine Science and Engineering Guangdong Laboratory (Zhuhai), Zhuhai, China. [9]These authors contributed equally: Fukai Liu, Xun Li. ✉e-mail: fliu@ouc.edu.cn; yiyongluo@ouc.edu.cn

thereafter (Supplementary Fig. 1a). Since AAs are known to be an important radiative forcing that cools the global climate and drives an increase in the surface water density over the North Atlantic[18,20,21], the AAs-induced AMOC decay in recent decades is presumably attributed to the decreased AAs emission over the adjacent North America and Europe[13,14,21,22]. In contrast, the increased AAs over the eastern hemisphere are anticipated to have a cooling effect, hypothetically strengthening the AMOC[17]. However, owing to their inhomogeneous spatial distributions[23–25], changes in Asian AAs result in dynamic impacts on large-scale atmospheric circulations and have far-reaching effects on remote regions. A recent study, in particular, has demonstrated that increased Asian AAs can shift the westerly jet equatorward in the northern hemipshere[26]. However, the influence of the increased Asian AAs on the AMOC and the underlying mechanisms remain unknown. Here, we find that far-field AAs over Asia weaken the AMOC despite local radiative cooling.

## Results

### Model set-up and experimental design

As discussed above, aerosol optical depth evolution is marked by a clear spatial contrast with a positive trend over the eastern hemisphere and a negative trend over the western hemisphere (Fig. 1a), known as the zonal shift pattern of AAs emission[17,27]. To investigate the climatic effects of changes in eastern and western AAs changes, we perform idealized radiative perturbation experiments[17,28,29] using a widely adopted fully-coupled climate model−the Community Earth System Model version 1 (CESM1) (see "CESM1 model" in Methods), known for its ability to represent the main physical processes involved in AMOC variability[3,30–32]. Prior to the perturbation runs, we first integrate a 500-yr control (CTRL) by forcing the CESM1 with GHGs, AAs, and solar insolation fixed at the year 2000 levels. To mimic the radiative effect of the aerosol forcing, we separately reduce solar insolation by 10% over East Asia and South Asia (red box in Fig. 1a) and North America and Europe (blue boxes in Fig. 1a), hereafter referred to as EAST and WEST

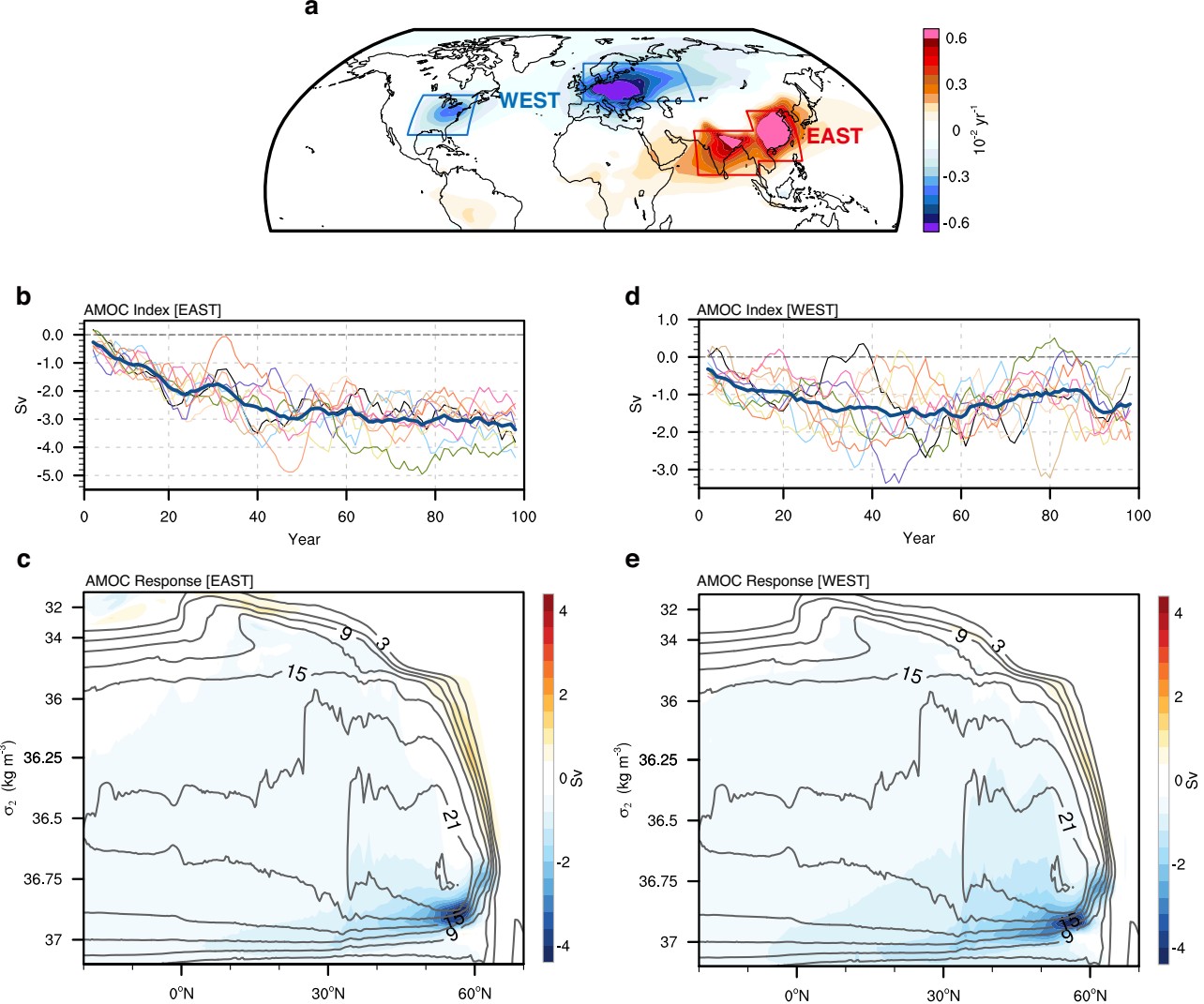

**Fig. 1 | Simulated slowdown of the Atlantic Meridional Overturning Circulation (AMOC) due to Asian anthropogenic aerosols (AAs) changes. a** Aerosol optical depth trends (unit: 0.01/year) between 1986–2014. The red box denotes the perturbed region in EAST simulations (South Asia box: 8–30°N, 65–100°E, East Asia box: 15–40°N, 100–125°E), and the blue boxes denote the perturbed region in WEST simulations (North America box: 28–48°N, 103–68°W, Europe box: 45–65°N, 0–70°E). **b** Ensemble-mean evolution of annual-mean AMOC index (unit: Sv) in EAST smoothed with a 5-yr running-mean filter, with the thin curves denoting each

member's anomalies. **c** AMOC streamfunction response (unit: Sv) in density coordinate in EAST averaged over model years 1-40, and superimposed black contours are the AMOC climatology in CTRL. **d, e** Same as **b, c**, but in WEST. The model simulated AMOC index in **b, d** is obtained as the maximum of streamfunction between latitudes 25° N–75° N and between depths of 500 and 2000 m. The radiative cooling resulting from increased Asian AAs paradoxically drives an AMOC slowdown.

simulations, respectively. Assuming linearity, the sign of the response in WEST is reversed to mimic the radiative heating effect of the aerosol decrease after the 1980s. The annual-mean net radiative forcing amounts to ~30 $Wm^{-2}$ in the perturbation experiments, which is three times larger in magnitude than the actual radiative forcing of the order 10 $Wm^{-2}$[17], for the purpose of generating a robust response signal.

According to paleoclimate proxy records and climate models, the AMOC exhibits decadal to multi-decadal variability[33–36]. To ensure that the simulated changes are not the result of internal variability, a total of ten 100-yr ensemble members are generated for each of the CTRL, EAST, and WEST experiments branching out from year 350 of CTRL, where each member is realized by adding a small random perturbation to the initial surface temperature in each realization. Since we focus on how increased Asian AAs and the associated radiative cooling affect the AMOC through exciting atmospheric teleconnections during the transient response stage, the main analyses shown in this study are based on the difference between years 1–40 of the ensemble perturbed runs and the corresponding 40 years of the ensemble CTRL. In addition, we provide an overview of the ensemble responses during the slow response phase averaged over years 51–100 in Supplementary Fig. 2, and the consistency between the results averaged over the first 40 years and the last 50 years suggests a short response timescale of the AMOC to changes in AAs.

## Increased Asian aerosols weaken the AMOC

It is somewhat surprising to see that the aerosol-induced cooling over Asia leads to a significant slowdown of the AMOC (Fig. 1b). This slowdown includes a rapid decrease of ~2.7 Sv (1 Sv=$10^6 m^3 s^{-1}$) within the first 40 years across all ensemble members (Fig. 1b, thin colored lines) and a gradual decline by ~0.6 Sv over the subsequent 60 years. This result demonstrates that the simulated decline of the AMOC is a robust climate response and not related to internal climate variability. Regarding the temporal evolution of the AMOC response in the WEST (Fig. 1d), as expected, the AMOC is weakened by ~1.4 Sv in the first four decades due to the warming effect of reduced AAs, and remains relatively stable around this value thereafter. Note that the magnitudes of both the transient and long-term responses of the AMOC in WEST are

smaller than those in EAST, suggesting that the remote Asian forcing exerts an even stronger influence on the AMOC than the adjacent forcing over North America and Europe.

The transient AMOC responses in density space (see "AMOC in density space" in Methods) in both EAST and WEST exhibit strikingly similar spatial patterns (Fig. 1c, e). The reduction occurs in the lower limb of AMOC, between 36.80 and 37.00 $kgm^{-3}$. This reduction signifies a decrease in the transport of equatorward-flowing deep water, indicating not only an overall weakening but also a shallower AMOC. The wind-driven subtropical cells in the light upper limb ($\sigma_2 < 33.5$ $kgm^{-3}$) appear to be rarely affected by the radiative forcing, regardless of whether it originates from the eastern or western hemisphere. This demonstrates that the primary impact of aerosol changes is concentrated on the lower branch of the AMOC.

As shown in the DAMIP single forcing experiments (Supplementary Fig. 1a), the rate of AAs-induced AMOC weakening after the 1990s ($-0.81$ Sv/decade during 1990–2014) is 1.4 times greater than the AMOC strengthening before the 1990s (0.56 Sv/decade during 1920–1990), despite the limited changes in global mean AAs concentration since the 1980s. Our model experiments suggest that the simulated acceleration in AMOC weakening after the 1990s is likely a combined result of AAs changes in both the eastern and western hemispheres. While the radiative heating caused by aerosol reduction over North America and Europe weakening the AMOC is not surprising, the mechanism behind the AMOC slowdown induced by remote radiative cooling over Asia remains puzzling. Furthermore, the remote forcing appears to have an even stronger influence on the AMOC than the adjacent forcing surrounding the North Atlantic. Next, we will delve into the mechanisms behind the AMOC slowdown in EAST.

## Circumglobal teleconnection excited by increased Asian Aerosols

The impact of Asian AAs-induced radiative cooling on the AMOC is conducted by exciting stationary Rossby wave patterns. Initially, the cooling effect leads to the suppression of local convection and the development of upper-level convergence over broad regions in India and China (Fig. 2a), inducing quasi-stationary atmospheric planetary

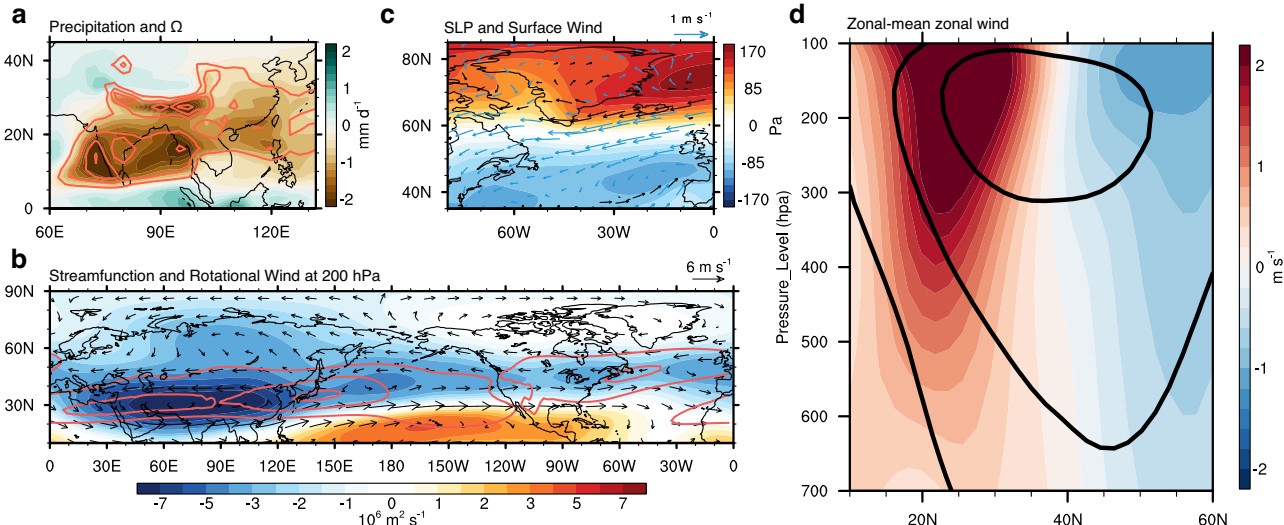

**Fig. 2 | Circumglobal atmospheric responses excited by increased Asian anthropogenic aerosols (AAs).** **a** Responses of precipitation (colors; unit:$mmday^{-1}$) and vertical velocity ($\Omega$) averaged over 200-1000 hPa (contour; unit: Pa/s) over South and East Asia. **b** Responses of 200-hPa streamfunction (colors; unit:$10^6 m^2 s^{-1}$) and rotational wind (vectors; unit:$ms^{-1}$) in the northern hemisphere, superimposed are the climatological 200-hPa zonal wind (contours; unit: unit: $ms^{-1}$). **c** Responses of sea level pressure (SLP; colors; unit: Pa) and surface

wind (vector; unit:) over the North Atlantic. Blue and black vectors denote anomalous winds that weaken and intensify the climatological winds. **d** Response of the zonal-mean zonal winds (colors; unit:) in the northern hemisphere, and superimposed is the corresponding climatology in CTRL. The increased Asian AAs weaken the westerlies over the subpolar North Atlantic by exciting circumglobal stationary Rossby waves in the mid-latitudes.

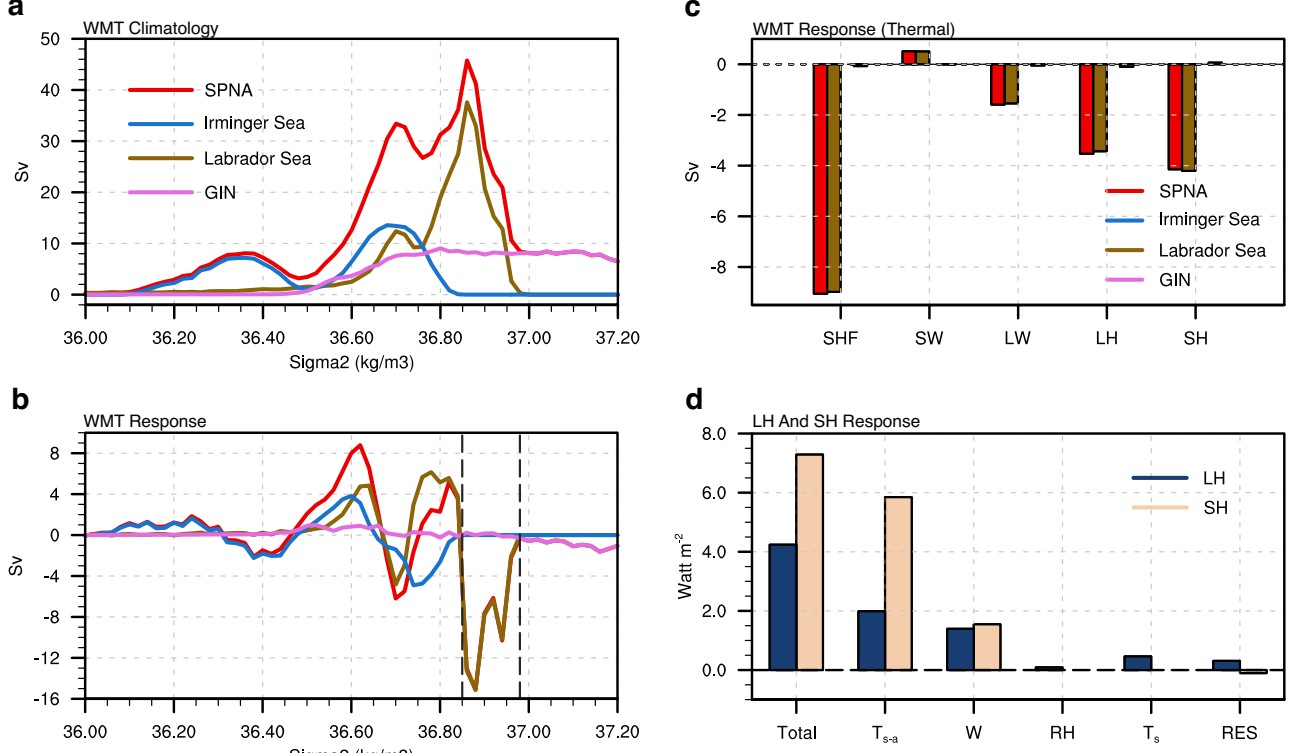

**Fig. 3 | Winter Water Mass Transformation (WMT) responses over the subpolar North Atlantic to increased Asian anthropogenic aerosols. a** Surface-forced WMT climatology (unit: Sv) over the subpolar North Atlantic (SPNA) and contributions from Irminger Sea, Labrador Sea, and Greenland-Iceland-Norwegian Seas (GIN). **b** Same as **a** but for the WMT response. **c** The thermal component of WMT response (SHF) averaged over the density range 36.85-36.98 (denoted by the dashed black lines in **b**), and its components due to shortwave radiation (SW), longwave radiation (LW), latent heat flux (LH) and sensible heat flux (SH).

**d** Responses of latent heat flux and sensible heat flux (unit: $Wm^{-2}$; positive downward) over the Labrador Sea, and their decomposition into contributions from air-sea temperature difference ($T_{s-a}$), surface wind speed (W), relative humidity (RH), SST ($T_s$), and a residual term (RES). The weakened westerlies suppress evaporation by reducing air-sea temperature difference, which hinders water mass transformation at the deep convection sites and thus causes the Atlantic Meridional Overturning Circulation slowdown.

wave trains propagating poleward and eastward (Fig. 2b), manifested as an equatorward shift of the westerly jet in the northern mid-latitudes (Fig. 2d). The associated streamfunction anomalies predominantly exhibit zonal wavenumber 4-5, characterized by a few prominent centers over the Tibetan Plateau, North Pacific, North America, and western Europe, which are aligned within the waveguide of the westerly jet stream. This anomalous pattern of atmospheric circulation mainly occurs within the jet stream waveguide, resembling the well-known circumglobal teleconnection[37,38]. In this case, however, the centers for East Asia and the Indian regions are merged due to the idealized configuration of the radiative forcing, covering western, central, and eastern Asia. The circumglobal response triggered by the reduction in Asian AAs persists throughout the year (Supplementary Fig. 3a–d), with its peak intensity occurring in summer, likely the result of the stronger convective feedback to external radiative forcing and hence a stronger diabatic wave source during the active Asian Monsoon season (i.e., summer) than winter (Supplementary Fig. 3e).

The temporal evolution of streamfunction anomalies along the mean path of the stationary wave shows that the teleconnection response emerges within a few months and stabilizes in the first few years (Supplementary Fig. 4). This suggests a relatively short timescale for the northern hemisphere atmospheric circulation to respond to changes in Asian AAs. On this time scale, the oceanic feedback to the atmospheric circulation is expected to have a secondary influence. The response of the atmospheric circulation is largely barotropic. Over the subpolar North Atlantic, the induced sea level pressure response is characterized by a year-round meridional dipole (Fig. 2c, colors and Supplementary Fig. 5), resembling the negative phase of the North

Atlantic Oscillation (NAO). Notably, there are positive pressure anomalies over Iceland and negative anomalies in lower latitudes. These sea level pressure anomalies correspond to a weakening and equatorward shift of the prevailing westerlies located to the south of Greenland (Fig. 2c, vectors). The equatorward shift of the westerly jet over the North Atlantic since the 1980s has been observed and simulated[23,39], and a recent study has pinpointed the primary role of increased Asian AAs in driving this shift in purposefully designed experiments[26], thus corroborating our findings. The next natural question is: How do these changes in atmospheric circulation weaken the AMOC?

**Suppressed westerly winds hinder the deep water formation**

Atmospheric circulation changes over the North Atlantic (especially North Atlantic Oscillation-related changes) have been proven important in transforming upper ocean waters into the densest class of the North Atlantic Deep Water[40–43]. To investigate the regional surface ocean drivers responsible for the reduced transport of lower limb waters (Fig. 1c), we conduct a winter-time surface water mass transformation (WMT) analysis (see "Water mass transformation" in Methods). Climatologically, the partitioning of the total WMT during winter into different regions varies significantly across different density classes (Fig. 3a). In particular, in the higher density range (>36.80 $kgm^{-3}$), the WMT primarily occurs in the Labrador Sea, indicating its important role in maintaining the transport of the lower limb waters. The Greenland-Iceland-Norwegian Seas contribute to the WMT in a density range greater than 37.00 $kgm^{-3}$, which deviates a lot from the maximum AMOC densities ($\sigma_2 = 36.73\ kgm^{-3}$; Fig. 1c, contours) and is

thus unlikely to contribute to the AMOC. These climatological results agree well with previous modeling studies[44,45]. Although the contribution of the Labrador Sea to the mean deep convection may be overestimated in climate models[46], recent studies suggested that long-term changes in the AMOC are still dominated by changes in deep convection in the Labrador Sea[45,47].

In response to the radiative cooling over Asia, a pronounced reduction of the winter WMT occurs in the subpolar North Atlantic (Fig. 3b). This reduction is centered at $\sigma_2 = 36.88\ kgm^{-3}$ and reaches a maximum value of 15.1 Sv, and the annual-mean reduction (5.6 Sv) matches the weakened AMOC in a similar density range. Among the four major deep convection sites, the Labrador Sea stands out as the dominant contributor to the WMT reduction, while the other basins have little contribution. In addition, the Labrador Sea experiences an enhancement of the WMT at lower density, with a peak value of 6.1 Sv at $\sigma_2 \sim 36.78\ kgm^{-3}$. Therefore, the WMT anomaly in the Labrador Sea manifests as an overall shift towards a lower density range, and the simulated AMOC slowdown can be entirely attributed to the reduction of the WMT in the Labrador Sea.

To investigate the driving mechanism behind the WMT reduction, we analyze the response of surface density flux in different basins, which measures the changes in surface buoyancy due to heat and freshwater exchanges between the atmosphere and the ocean (see "Water mass transformation" in Methods)[48]. The reduced WMT almost entirely resulted from the suppressed heat loss in the surface Labrador Sea (Fig. 3c, also see Supplementary Fig. 6 for the negligible haline contribution), resulting in a more buoyant surface ocean that hinders the transformation of lighter upper waters into denser waters. By quantifying the thermally-driven WMT anomaly associated with different surface heat flux components, we find that the WMT reduction is mainly driven by reduced turbulent heat loss (also see Supplementary Fig. 7a-b for their spatial patterns). Both latent and sensible heat fluxes contribute to the reduction in the Labrador Sea, with the former contributing −3.5 Sv and the latter −4.1 Sv.

The reduced turbulent heat flux over the Labrador Sea is primarily driven by the suppressed westerlies over the subpolar North Atlantic. The total turbulent heat flux changes (11.5 $Wm^{-2}$) can be linked to changes in SST, surface wind speed, surface relative humidity, and air-sea temperature difference (see "Turbulent flux decomposition" in Methods). The largest contribution arises from the reduction in the air-sea temperature difference (7.8 $Wm^{-2}$; Fig. 3d, second blue and yellow bars), which in turn can be attributed to the weakened surface westerlies south of Greenland (Fig. 2c). Specifically, the suppressed surface wind reduces the transport of cold air from the upstream North American continent to the deep convection site, generating an anomalously warmer boundary air than the surface ocean, and thus inducing an anomalously zonal warm temperature advection over a substantial region of the Labrador Sea (Supplementary Fig. 7c–f). Moreover, the reduction in wind speed also acts to warm the surface ocean by suppressing heat loss from latent and sensible heat fluxes (2.9 $Wm^{-2}$; Fig. 3d, third blue and yellow bars). Therefore, the suppressed westerlies lead to a decrease in both air-sea temperature difference and surface wind speed, collectively hindering turbulent heat loss and impeding the deep water formation in the Labrador Sea (Fig. 4).

The rate of AMOC weakening slows down substantially as time progresses into the slow response stage (Fig. 1b). Given that the AMOC is responsible for most of the heat transport in the Atlantic basin, its weakening results in reduced poleward oceanic heat transport (Supplementary Fig. 8a). As a result, anomalously cold water (Supplementary Fig. 8b) propagates downstream along the cyclonic subpolar gyre, eventually reaching the Labrador Sea. This process acts as a negative feedback, balancing the surface heat gain in the Labrador Sea and preventing further reduction of deep water formation. Consequently, this self-limited process helps slow down the weakening of the AMOC.

## Additional multi-model evidence confirming the Asian AAs-AMOC link

The above responses in atmospheric teleconnection and surface-forced WMT identified for weakening the AMOC are behaviors expected from fundamental physical principles, but the results are based solely on the CESM1 model. It remains to be seen whether the main conclusion holds in different model settings. Therefore, we perform a comprehensive analysis including multi-model results and large-ensemble simulations to test the robustness of our findings.

We use simulations from 10 models participating in the Aerosol Chemistry Model Intercomparison Project (AerChemMIP)[49], which is a subset of Coupled Model Intercomparison Project Phase 6 (CMIP6) dedicated to understanding the fast responses of the atmosphere to aerosol changes (see "CMIP6 and CESM2-SF-LE simulations" in Methods). We make use of two sets of AerChemMIP simulations, piClim-control and piClim-SO2, each spanning 30 years with fixed climatological mean SST and sea ice conditions corresponding to the pre-industrial levels. The only difference between these two sets of simulations is the emissions of anthropogenic aerosol precursors of $SO_2$: while piClim-control employs preindustrial emissions fixed at 1850 levels, piClim-SO2 adopts emissions fixed at 2014 levels. Notably, due to the implementation of the Clean Air Act in Europe and North America, $SO_2$ emissions in the western hemisphere have already decreased significantly by 2014, making Asian emissions the predominant forcing in the piClim-SO2 simulations (Supplementary Fig. 9a, contours). Therefore, this 2014 sulfate forcing operates as a comparable agent to that used in our EAST simulations (Fig. 1a), albeit being more concentrated in East Asia.

To assess the transient responses of atmospheric circulation to Asian AAs, we calculate the difference between the piClim-SO2 ensemble and the piClim-control ensemble for years 1–30. Consistently, the AerChemMIP simulations show a similar atmospheric response pattern compared to our EAST simulations (Supplementary Fig. 9a, colors), especially the circumglobal teleconnection structure in the northern midlatitudes. The anomalous low-pressure centers of the teleconnection patterns in the AerChemMIP simulations are slightly shifted compared to those in our EAST simulations (Fig. 2b), due possibly to the differences in the distribution of aerosol forcing. Nevertheless, the equatorward shift of the westerly jet and the negative NAO-like response are evident (Supplementary Fig. 9b), accompanied by suppressed westerlies over the Labrador Sea. These agreements between the results of AerChemMIP and our CESM1 simulations provide support for the findings from our CESM1 simulations, adding confidence in the robustness of the atmospheric teleconnections excited by increased Asian AAs. However, the response of the AMOC is elusive in the AerChemMIP setting due to the absence of an active ocean model.

Recognizing the limitations of AerChemMIP and the incapability of distinguishing the effects of aerosol changes between the eastern and western hemispheres in DAMIP experiments, we turn to the CESM2 Single Forcing Large Ensemble (CESM2-SF-LE). This ensemble extends its 15-member AAs-only simulations to the year 2050 under the shared socioeconomic pathways 3-7.0 scenario, referred to as SSP370-AAs simulations. In this scenario, Asian AAs emissions are projected to continue to increase in the future, while AAs in the western hemisphere show little change as they have already decreased to low levels (Supplementary Fig. 10b, contours). Consequently, the increased Asian AAs and the associated radiative cooling emerge as a dominant forcing during the period 2015 to 2050. This provides an analog to our EAST case (Fig. 1a) and adds some model diversity to our analysis.

Consistent with our EAST case, the SSP370-AAs simulations also show a circumglobal teleconnection structure and a negative NAO-like trend in the atmospheric circulation (Supplementary Fig. 10b–d). In addition, the weakened westerlies over the Labrador Sea lead to a

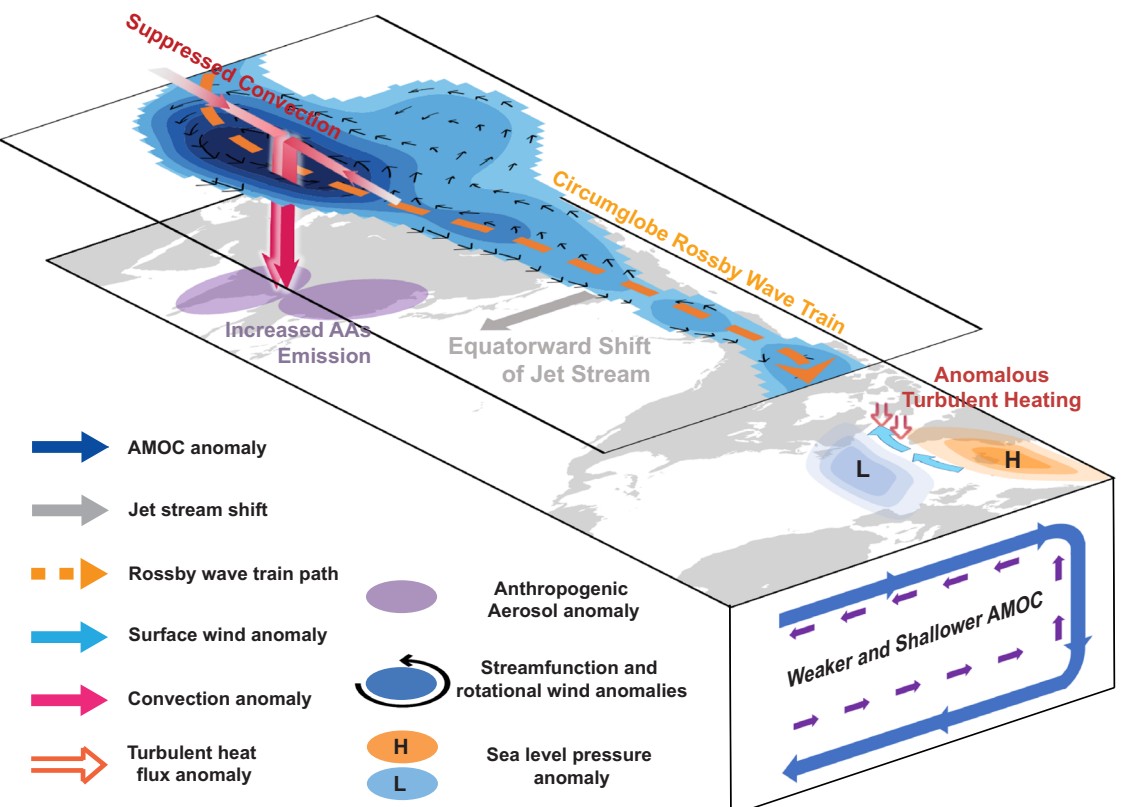

**Fig. 4 | Schematic of the weakened Atlantic Meridional Overturning Circulation (AMOC) due to increased anthropogenic aerosols (AAs) in Asia.** Highlighted are the responses in the surface, upper troposphere, and ocean in EAST. Increased AAs emissions over South and East Asia (purple shading) reduce the solar insolation and suppress local convection (red arrows), igniting a circumglobal Rossby wave train (blue shading and black arrows) propagating eastward and poleward and an equatorward shift of jet stream (gray arrows). The barotropic atmospheric circulation changes generate a negative North Atlantic Oscillation-like sea level pressure anomaly in the subpolar North Atlantic (orange high pressure center and blue low pressure center), suppressing the prevailing westerlies south of Greenland (light blue arrows). Through reducing air-sea temperature difference and wind speed, the wind changes inject anomalous turbulent heat into the surface Labrador Sea (red arrows), inhibiting the production of dense water masses that make up the southward flowing limb of the AMOC, and thus resulting in a weaker and shallower AMOC (blue shading in the ocean).

reduction in heat loss through changes in turbulent fluxes, ultimately resulting in a weakening of the AMOC. In particular, the AMOC weakening also exhibits two distinct timescales (Supplementary Fig. 10a): an initial rapid decline followed by a relatively slower rate of change. Such an alignment of these results suggests that the relationship between Asian AAs and the AMOC is not unique to CESM1, lending more support to our conclusion that the increased Asian AAs can weaken the AMOC.

## Discussion

In this work, we find that the increased Asian AAs, despite causing a radiative cooling, drive a weakening of the AMOC. This finding is corroborated by the analysis of AerChemMIP and CESM2-SF-LE experiments with similar AA forcings. The pathway of this effect of Asian AAs starts with suppressed convection over South and East Asia, which excites a Rossby wave train propagating eastward and poleward. The barotropic atmospheric circulation anomaly generates a negative North Atlantic Oscillation-like response and suppresses westerly winds south of Greenland, which hinders the transformation of upper ocean waters in the Labrador Sea via turbulent heat fluxes and ultimately leads to the slowdown of the AMOC. A recent study[17] employing a similar experimental setting has reported a contrasting result that increased Asian AAs could enhance the AMOC. The forcing in their experiment is placed over Siberia, and the induced cooling is potentially advected downstream to the subpolar North Atlantic by the mean westerlies, resulting in surface ocean cooling and thus AMOC strengthening. By contrast, in our study, the aerosol forcing located at

lower latitudes affects the AMOC by inducing a circumglobal Rossby wave train. The contrasting result underscores high sensitivity to the location of the forcing[29]. Our discovery of a link between increased Asian AAs and a weakened AMOC suggests that the combined effects of increased AAs in Asia and decreased AAs in North America and West Europe, which represent the most consequential changes in aerosol emissions over the past few decades, conspire to contribute to the slowdown of the AMOC, with the decreased Asian AAs having a larger effect.

## Methods

### CMIP6 and CESM2-SF-LE simulations

To estimate the impact of different anthropogenic forcing on the AMOC during the historical period, we use outputs of AMOC and aerosol optical depth at 550 nm from the Detection and Attribution Model Intercomparison Project (DAMIP)[19] based on data availability. The DAMIP simulations include scenarios with GHGs only (Hist-GHGs), AAs only (Hist-AAs) simulations and cover the period of 1850-2014.

To validate the contributions of increased Asian AAs on atmospheric circulation changes, we use Aerosol Chemistry Model Intercomparison Project (AerChemMIP)[49] outputs of aerosol optical depth at 550 nm, zonal and meridional wind velocity, sea level pressure data from the piClim-control and piClim-SO2 simulations. These simulations are integrated for 30 years and are identical in all ways, except that piClim-control employs preindustrial $SO_2$ emissions, whereas piClim-SO2 adopts present-day $SO_2$ emissions. Only the first member of each model is used to ensure equal weight in the multi-model

ensemble mean analysis. A detailed summary of these CMIP6 simulations can be found in Supplementary Table 1.

In addition, to further validate the contributions of increased Asian AAs on AMOC weakening, we use CESM2 Single Forcing Large Ensemble (CESM2-SF-LE)[50] outputs from 15 ensemble members of the AAs-only simulations. All simulations use the same AAs forcing based on the SSP370 scenario from 2015 to 2050. Ensemble members are generated by round-off level differences in their initial air temperature fields.

## CESM1 model

The model we use is the Community Earth System Model version 1 (CESM1)[51], with a gx1v6 global configuration that comprises an atmosphere (Community Atmosphere Model version 5), a land (Community Land Model version 4), an ocean (Parallel Ocean Program version 2), and a sea ice (Community Ice CodE) component model. The atmosphere and land models run on a nominal 2° horizontal grid, and the sea ice and ocean models use an irregular horizontal grid of a nominal 1° resolution, but telescoped meridionally to ~35–50 km around Greenland and the Labrador Sea ("f19_gx1v6" horizontal resolution). Vertically, the atmosphere model discretized on 30 uneven vertical levels, and the ocean model has 60 uneven levels with the thickness varying from 10 m near the surface to 250 m near the bottom.

## AMOC in density space

We present the spatial pattern of AMOC in density space, as it facilitates relating AMOC changes to variations in water mass transformation in the surface ocean. AMOC in density is calculated as[45]:

$$\text{AMOC}(\sigma_2, y, t) = -\int_{x_W}^{x_E} \int_{-h}^{z} \upsilon(x, y, z, t) dz dx, \quad (1)$$

where $\sigma_2$ is the potential density referenced to 2000 m, y is the latitude, t is time, $x_E$ and $x_W$ are the western and eastern longitudinal limits of the Atlantic basin, $\upsilon$ is the meridional velocity, and $h$ is the bottom depth.

## Water mass transformation

Because of the dominance of surface buoyancy exchange in driving the density-space overturning in the deep convection sites[47], surface-forced WMT analysis serves as a powerful tool for studying deep water formation and AMOC variation in both observations and models. Following the methodology in previous studies[47,48,52], the surface density flux (SDF) can be calculated as

$$\text{SDF} = -\alpha \frac{Q}{C_p} - \beta \frac{F}{1-S} S, \quad (2)$$

where the first term represents the thermal contribution related to surface heat fluxes (Q) and the second term is the haline contribution associated with freshwater fluxes (F). The thermal component includes contributions from latent and sensible heat fluxes, shortwave, and longwave radiations. The haline component mainly comprises precipitation, evaporation, runoff, melting flux, and salt flux due to sea ice formation/melt. $\alpha$ and $\beta$ are the thermal expansion and haline contraction coefficients as functions of time and location, $C_p$ is the specific heat capacity of seawater, and S is the sea surface salinity. To compute the surface-forced WMT as a function of density, the SDF is integrated over all surface density outcrop regions in each density bin:

$$WMT(\sigma) = \frac{1}{\Delta\sigma} \int\int_{\sigma}^{\sigma+\Delta\sigma} SDF dA, \quad (3)$$

Where $\sigma$ is the potential density referenced to 2000 m. The above analysis is based on monthly mean model outputs.

## Turbulent flux decomposition

The latent heat flux response can be decomposed to contributions from the air-sea surface temperature difference ($\Delta T$), surface wind speed (W), relative humidity (RH), and SST ($T_s$)[53]:

$$Q_{LH}^{\Delta T'} = \bar{Q}_{LH} \frac{\beta \overline{RH} e^{-\beta\Delta T}}{1 - \overline{RH} e^{-\beta\Delta T}} \Delta T', \quad (4)$$

$$Q_{LH}^{W'} = \bar{Q}_{LH} \frac{W'}{\overline{W}}, \quad (5)$$

$$Q_{LH}^{RH'} = -\bar{Q}_{LH} \frac{e^{-\beta\Delta T}}{1 - \overline{RH} e^{-\beta\Delta T}} RH', \quad (6)$$

$$Q_{LH}^{o'} = \beta \bar{Q}_{LH} T_s', \quad (7)$$

where $\beta = L_v/(R_v T_S^2)$ with $L_v$ the latent heat of condensation and $R_v$ the ideal gas constant for water vapor. The overbar represents the climatological state in CTRL, and the prime denotes the deviation from the climatological state. Similarly, the sensible heat flux response can be decomposed to contributions from the air-sea surface temperature difference and surface wind speed:

$$Q_{SH}^{\Delta T'} = \bar{Q}_{SH} \frac{\Delta T'}{\overline{\Delta T}}, \quad (8)$$

$$Q_{SH}^{W'} = \bar{Q}_{SH} \frac{W'}{\overline{W}}. \quad (9)$$

## Data availability

The CMIP6 data is publicly available at: https://esgf-node.llnl.gov/. The CESM2-SF-LE data is publicly available at: https://www.cesm.ucar.edu/community-projects/lens. The RAPID data is publicly available at: https://rapid.ac.uk/rapidmoc/. The CESM1 simulation data to support the analysis is available at https://doi.org/10.5281/zenodo.10374334.

## Code availability

The NCAR Command Language (NCL v6.5.0) is used for plotting. Codes to reproduce the figures are available at https://doi.org/10.5281/zenodo.10374334.

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

## Acknowledgements
This work was supported by the National Natural Science Foundation of China (No. 42230405), the Science and Technology Innovation Foundation of Laoshan Laboratory (No. LSKJ202202401), and Fundamental Research Funds for the Central Universities (No. 202341016). F.L. is supported by the "Youth Innovation Team Program" Team in Colleges and Universities of Shandong Province (No. 2022KJ042). J.L. is supported by Office of Science, U.S. Department of Energy (DOE) Biological and Environmental Research as part of the Regional and Global Model Analysis program area.

## Author contributions
F.L. conceived the initial idea, interpreted the results, and wrote the initial manuscript; X.L. participated in preliminary analysis and plotted the figures; Y.L. led the research and improved the manuscript; F.L. and X.L. performed the CESM1 experiments; W.C., J.L., and X.T.Z. contributed to interpreting the results and improving the manuscript. S.M.K., H.W, and L.Z. contributed to discussing the results and the manuscript.

## Competing interests
The authors declare no competing interests.
