## [Peer Review File · Nature Communications]

Increased Asian aerosols drive a slowdown of Atlantic Meridional Overturning CirculationREVIEWER COMMENTS

Reviewer #1 (Remarks to the Author):

Review of "Increased Asian aerosols drive an unexpected slowdown of Atlantic Meridional Overturning Circulation" by Liu et.al.

This study investigates the role of the increasing AA emissions over Asia on the AMOC changes, especially for a decline in AMOC through designed experiments using the CESM climate model. The main argument of the study is that the increasing AA over Asia contributes to a decline in AMOC after 1990s. The results from the experiments are clear and supportive of the argument drawn from the analysis. However, in my view, the authors need to show more rigor in their analysis with some further clarifications to establish the novelty of their results according to the standard of this journal.

Main Comments:

1. I am not sure why the authors claim this response "unexpected". I find this phrasing unnecessary. Because I don't find any clear evidence of expecting anything else to be happening from AA aerosol emissions from Asia. There could be other hypothesis which their results do not support. But a hypothesis described in one study is not enough to set an expectation. This is the first study looking at this aspect systematically and finding a clear connection from the AA emissions in Asia to AMOC changes. Moreover, I can't also see any basis to consider the response from Asian AA emissions to be in similar nature to the European or North American AA emissions. Because the AA emission over Asia is remote in nature and hence its connection to AMOC can't be considered in similar lines. I would recommend to omit this "unexpected" term or else to provide clear evidence of previous study looking into the very same response and finding a totally opposite outcome.

2. The other glaring major issue is that the study is based on sensitivity experiments conducted on a single model CESM1. There is no analysis conducted on finding the relation of the AA changes in the Asia and AMOC strength change in observations. Because the observed behaviors of AMOC most probably be very different from the models such as CESM1 (see Lozier et.al. 2019). The main influence region for AMOC from Asia AA emissions in the model is over Labrador Sea. Recent observational studies are showing that the observed AMOC sensitivity to Labrador Sea deep convection changes maybe minor compared to what some standard coupled climate models depicts. This means that though the analysis shows a valid way of Asian AA emission influence to AMOC through Labrador Sea water mass transformation changes, such change may not be happening in reality due to differences in sensitivities in the observed world from a model world. A more rigorous analysis on observed changes and its connection to observed AMOC changes is necessary to give this study a stronger foundation. The authors can look at observed year to year AA changes over Asia and investigate if there is any connection to the observed AMOC changes. Further, the same could be investigated for the teleconnection pathways. Do we see a similar teleconnection pathway from the Asia in the observed upper-level winds? There must be some signal present in the observational geopotential height, winds and stream function field in association with observed AA changes over Asia, if a similar mechanism is at play. Do we also find a

negative NAO like response connection to Asian AA changed over the North Atlantic region. Previous studies could have provided some hint on this, though I think it is essential to provide a clear picture of such observational association in this manuscript.

3. It would be also useful and highly confirmatory to show if CMIP6 models altogether show a same connection of AA emissions changes over Asia and AMOC. Among the CMIP6 models, when there are different future AA emission levels or aerosol radiative effect (under the same scenario) over East Asia, do they show any correlation with their respective AMOC changes? The goal is if the authors could show somehow the model independency of the main findings and a similar connection from AA emissions over Asia to AMOC across multiple models.

Specific comments:

Line 76: what do you mean by “present day level”?

Line 77: why 10% ?

Line 80: here the response is reversed, which means the authors assume a linear nature of the response. This has to be stated and if the authors could provide some evidence behind such an assumption that would be even better.

Line 85: why only 10 years of simulations are performed? This is quite small and unusual. I understand that the response time for aerosol would be small but a time of emergence analysis on the AMOC strength should be done considering its variability in the model to determine the minimum duration of the run to find a discernible forced signal from internal variability. I am okay with the claims made qualitatively from this limited length data, though the quantification could be different and could show a further robust estimate if the runs are performed for longer times. I would urge the authors to do this if there is no major computational constraint or put some strong reasoning to perform such short simulations. It is not ideal to set such experimental precedence without any strong reasoning.

Line 98: Why is there a rapid decrease in the first two decades? Could the authors give a reason/hypothesis for this?

Line 100: Similarly, why is there a gradual decline afterwards?

Line 108-111: Are these separate experiments performed with only South Asian AA and East Asian AA? The description of such experiments should be clearly stated then and the supplementary figure 3 also do not provide the images of two separate experiments. Hence it remains unclear what the authors have done.

Line 145: please remove “greatly”. And there are key differences in the CGT pattern and the CGT-like pattern that the authors find in this experiment. That has been pointed out too. I am curious if the authors could show such resembling pattern in observations with respect to observed Asian AA variations.

Line 150: is there an associated seasonal dependency in the AMOC response to Asian AA?

Figure 2c : in this figure the location of the mean westerlies in contours must be shown.

Line 165-167: here the authors claim that previous studies from CMIP5 and CMIP6 models already corroborate the link of the equatorward shift of the jet and AA emission over Asia. In a similar line, could the authors also link the changes in AA over Asia and AMOC changes in those CMIP5 and CMIP6 models?

Line 180: these are model specific behaviour of the AMOC and authors must be careful to generalise such things. Could the author confirm the same in observations? Lozier et.al. suggest that observed AMOC behaves in a different fashion than what the models show and AMOC is more sensitive to the changes in the eastern side of subpolar North Atlantic than the western side over Labrador Sea. Hence this potential difference in observed and model behaviour and its effect on the results needs to be clarified.

Line 214: could the authors show the changes in horizontal (zonal and meridional) temperature advection to confirm that the less cold air is getting advected over the Labrador Sea from North America? This analysis would complete the picture by clearly showing anomalous horizontal warm air advection.

Line 470: cures  curves

M. S. Lozier et al. A sea change in our view of overturning in the subpolar North Atlantic. Science 363,516-521(2019). DOI:10.1126/science.aau6592

Reviewer #2 (Remarks to the Author):

Key Results

Recent work has highlighted how aerosols over the North Atlantic region can strengthen AMOC by cooling the deep water formation region, which strengthens meridional density gradients driving AMOC. This study finds a new and potentially important contrary effect: aerosols over South Asia and East Asia can weaken AMOC by shifting the Atlantic Westerly winds southward, decreasing heat loss and warming the deep water formation regions. Heat budget analysis makes it clear that the main mechanism is the reduction in transport of cold continental air over the Labrador Sea, which reduces ocean heat loss, cooling, and density increase there.

Validity, Data and Methodology

The authors themselves point out what I see as the main question mark about the validity of the result, which is that climate models such as theirs do not do a good job capturing the overflow component (from the Nordic Seas) of deep water formation and hence may over-emphasize the role of the Labrador Sea. Since this is a very challenging feature to model correctly, it is a shortcoming of virtually all studies of AMOC, and should not hinder publication.

Significance

In recent decades, the largest aerosol increases have been in Asia, and this study is the first to show a clear link between these increases and possible weakening of AMOC. Therefore it is a highly important study. The fact that there has been increasing discussion of weakening of AMOC, presumably due to the effects of anthropogenic global heating, will heighten interest in this paper because it suggests an alternative explanation for dropping AMOC strength.

Clarity & Context/ Suggested Improvements

The paper is very well written. I have some comments on what is written about the context which also has some bearing on how the paper presents the significance of these findings.

1) Previous work on Asian Aerosols and teleconnections to N Atl

The last paragraph before the Discussion section discusses prior work, and it (or at least the first part of it, before the discussion of discrepancy with Kang et al and new results) should be placed at the end of the Introduction section. I assume the motivation for putting it later was that the paragraph is motivated by this paper's results that the main influence on AMOC of the Asian AA is through a shift of the westerlies. However, the paragraph (if placed in Intro) would make sense to a reader since it talks about effects of climate perturbations in Asia on the North Atlantic region, which is plausibly relevant to the topic at hand.

One sentence in that paragraph makes me think that this paper should summarize its results a little differently. The sentence says "experiments forced with ... aerosol emissions over the eastern hemisphere... has shown a weakening of the AMOC... providing direct support for our finding." If that were true, doesn't the existence an earlier paper with similar results call into question the novelty of the finding? I have not thoroughly read the paper in question (Diao et al, 2021), but as far as I can tell, Diao et al find the Asian aerosol effect on moving the Westerlies southward over the Atlantic, but not the further effect on the AMOC.

Therefore, the abstract and Intro should be rephrased a bit to make it clear that this study finds that a previously-discovered shift in Westerlies makes the Asian AA have the effect of decreasing AMOC.

2) Are the new results surprising?

This is more subjective, but I don't think the result is quite as surprising as they portray. The strength of the AMOC depends less on density than on density differences between the northern North Atlantic and

other parts of the Ocean (see Klinger and Haine, Ocean Circulation in Three Dimensions, Chapter 9). Thus, the fact that cooling the North Atlantic with aerosols strengthens AMOC does not imply that cooling a different latitude and longitude with aerosols should have the same affect. The Kang et al (2021) study cited here does suggest opposing AMOC increase rather than decrease, so maybe the real question this study is raising is not “what does Asian AA forcing do to AMOC?” but “what does Asian AA forcing at lower latitudes do to AMOC?”

3) Some minor suggestions

I recommend changing name of last section to "Discussion and Conclusions" and moving the comparison to earlier studies (end of last paragraph before discussion) to this section.

We thank the reviewers for their insightful comments and detailed suggestions, which are helpful in improving the manuscript. We have conducted further analysis with observations, CMIP6, and CESM2 large ensemble results to confirm that the link between increased Asian Anthropogenic Aerosols (AAs) and the AMOC is not unique to CESM1, the model used in this study. In addition, we have extended all 30 simulations (EAST, WEST, and CTRL) for an additional 80 years to validate the results of our study. We also thank both reviewers for their constructive comments that motivated us to write more meticulously. In the following, we provide a point-by-point response to both reviewers.

Note that major changes are highlighted in red in the revised text.

Reviewer #1 (Remarks to the Author):

Review of “Increased Asian aerosols drive an unexpected slowdown of Atlantic Meridional Overturning Circulation” by Liu et.al.

This study investigates the role of the increasing AA emissions over Asia on the AMOC changes, especially for a decline in AMOC through designed experiments using the CESM climate model. The main argument of the study is that the increasing AA over Asia contributes to a decline in AMOC after 1990s. The results from the experiments are clear and supportive of the argument drawn from the analysis. However, in my view, the authors need to show more rigor in their analysis with some further clarifications to establish the novelty of their results according to the standard of this journal.

Response: We appreciate your perceptive comments, which have helped us improve the manuscript substantially. As you can see from the revised version of our manuscript, considerable efforts have been made to address your comments, particularly in the following areas:

- (1) **Extension of Simulations:** We have extended all 30 simulations by an additional 80 years to further validate our results.
- (2) **Multi-Model Validation:** To demonstrate that our results are not model-dependent, we have analyzed a large number of simulations from two sets of AerChemMIP and one set of CESM2-LE, providing a multi-model validation of the findings from our CESM1 results.
- (3) **Regression Analyses:** We have conducted regression analyses using atmospheric reanalysis to investigate the relationship between changes in Asian AAs and atmospheric circulations.
- (4) **References:** We have incorporated references that examine long-term AMOC changes in high-resolution models, particularly emphasizing the driving role of the Labrador Sea. This contextualizes our work within the existing literature.
- (5) **Language Refinement:** We have refined the language, especially in the Introduction section, to highlight the novelty of this study without implying prior expectations.

Below are our point-by-point responses to your comments.

Main Comments:

1. I am not sure why the authors claim this response “unexpected”. I find this phrasing unnecessary. Because I don’t find any clear evidence of expecting anything else to be happening from AA aerosol emissions from Asia. There could be other hypothesis which their results do not support. But a hypothesis described in one study is not enough to set an expectation. This is the first study looking at this aspect systematically and finding a clear connection from the AA emissions in Asia to AMOC changes. Moreover, I can’t also see any basis to consider the response from Asian AA emissions to be in similar nature to the European or North American AA emissions. Because the AA emission over Asia is remote in nature and hence its connection to AMOC can’t be considered in similar lines. I would recommend to omit this “unexpected” term or else to provide clear evidence of previous study looking into the very same response and finding a totally opposite outcome.

Response: Thank you for your comment. Here we systematically investigate the connection between Asian AA emissions and AMOC changes. We called our result "unexpected" based on a single previous study (Kang et al. 2021) that seems to imply a strengthened AMOC in response to increased Asian AAs.

Following your suggestion, we have removed the word "unexpected". We have also rephrased the relevant sentences to better emphasize the novelty and uniqueness of our findings, without implying prior expectations.

Comment 2. The other glaring major issue is that the study is based on sensitivity experiments conducted on a single model CESM1. There is no analysis conducted on finding the relation of the AA changes in the Asia and AMOC strength change in observations. Because the observed behaviors of AMOC most probably be very different from the models such as CESM1 (see Lozier et.al. 2019). The main influence region for AMOC from Asia AA emissions in the model is over Labrador Sea. Recent observational studies are showing that the observed AMOC sensitivity to Labrador Sea deep convection changes maybe minor compared to what some standard coupled climate models depicts. This means that though the analysis shows a valid way of Asian AA emission influence to AMOC through Labrador Sea water mass transformation changes, such change may not be happening in reality due to differences in sensitivities in the observed world from a model world.

Response: Thank you for raising the issue. As you pointed out, the observation-based studies have indicated a limited role of the Labrador Sea in the mean deep convection and AMOC. However, recent studies have demonstrated that the Labrador Sea still plays a dominant role in long-term AMOC changes despite its weak mean deep convection and overturning in the mean state. For example, Oldenburg et al. (2021) compared the locations of deep convection in different low-resolution models and found that mean deep convection sites vary considerably between models,

yet the Labrador Sea drives the low-frequency AMOC variability, even in models where the time-mean deep convection occurs in the Irminger Sea or the Norwegian Sea. This result holds in high-resolution models (Yeager et al. 2021; Oldenburg et al. 2022), which reproduce the weak mean overturning along the OSNAP west line and can be seen as an analog to observations. These above findings reconcile the observation- and model-based perspectives on the subpolar origins of the AMOC: although the Labrador Sea plays a secondary role in driving the time-mean AMOC, it is still the most important driver of the long-term changes in the deep convection and the AMOC.

While we acknowledge that model simulations may not perfectly represent the observed AMOC, our study provides a starting point for investigating the influence of Asian AAs on the AMOC. We hope that our findings will stimulate research in model-based and observational studies.

A related discussion regarding the role of the Labrador Sea has been added to the revised text (Lines 186-188), and relevant papers (Lozier et al. 2019; Yeager et al. 2021; Oldenburg et al. 2022) have been cited in the revision.

References

- Lozier, M. S., F. Li, S. Bacon, F. Bahr, A. S. Bower, S. A. Cunningham, M. F. De Jong, et al. 2019. “A Sea Change in Our View of Overturning in the Subpolar North Atlantic.” *Science* 363 (6426): 516–21. <https://doi.org/10.1126/science.aau6592>.
- Oldenburg, Dylan, Robert C.J. Wills, Kyle C. Armour, Luanne Thompson, and Laura C. Jackson. 2021. “Mechanisms of Low-Frequency Variability in North Atlantic Ocean Heat Transport and AMOC.” *Journal of Climate* 34 (12): 4733–55. <https://doi.org/10.1175/JCLI-D-20-0614.1>.
- Oldenburg, Dylan, Robert C.J. Wills, Kyle C. Armour, and Lu Anne Thompson. 2022. “Resolution Dependence of Atmosphere–Ocean Interactions and Water Mass Transformation in the North Atlantic.” *Journal of Geophysical Research: Oceans* 127 (4): 1–18. <https://doi.org/10.1029/2021JC018102>.
- Yeager, Stephen, Fred Castruccio, Ping Chang, Gokhan Danabasoglu, Elizabeth Maroon, Justin Small, Hong Wang, Lixin Wu, and Shaoqing Zhang. 2021. “An Outsized Role for the Labrador Sea in the Multidecadal Variability of the Atlantic Overturning Circulation.” *Science Advances* 7 (41). <https://doi.org/10.1126/sciadv.abh3592>.

A more rigorous analysis on observed changes and its connection to observed AMOC changes is necessary to give this study a stronger foundation. The authors can look at observed year to year AA changes over Asia and investigate if there is any connection to the observed AMOC changes. Further, the same could be investigated for the teleconnection pathways. Do we see a similar teleconnection pathway from the Asia in the observed upper-level winds? There must be some signal present in the observational geopotential height, winds and stream function field in association with observed AA changes over Asia, if a similar mechanism is at play. Do we also

find a negative NAO like response connection to Asian AA changed over the North Atlantic region. Previous studies could have provided some hint on this, though I think it is essential to provide a clear picture of such observational association in this manuscript.

Response: Thank you for your insightful comments and we agree on the importance of observational evidence to strengthen the foundation of our findings. Following your suggestion, we have performed several analyses to explore the observational connections between changes in Asian AAs and atmospheric circulations.

First, we conducted a regression analysis using an atmospheric reanalysis dataset Modern-Era Retrospective analysis for Research and Applications, Version 2 (MERRA2), to examine the connection between changes in Asian AAs and atmospheric circulation in the real world. MERRA2 was chosen because it provides aerosol optical depth data. However, we encountered challenges in distinguishing the individual impacts of AAs and greenhouse gases (GHGs) due to the relatively smooth, long-term increase in both AAs and GHGs over the years, as shown in Fig. R1a,d. These commonalities resulted in the generation of very similar spatial patterns in the regression analyses for both GHGs and AAs (Fig. R1b,c,e,f), complicating the task of separating their distinct effects on atmospheric circulations.

We have performed an EOF analysis to identify the fingerprints in the atmospheric circulation associated with the Asian AAs changes. Figure R2 shows the first two empirical orthogonal functions (EOFs) modes of 200-hPa geopotential height in the northern midlatitudes, of which the variance fractions are 41.5% and 14.3%, respectively. The first EOF mode (Fig. R2a) and the corresponding principal component (PC) time series (Fig. R2b) indicate a monotonic poleward shift of the subtropical jet over the past few decades, primarily attributed to increased GHGs (Lu et al. 2010; Fu et al. 2011). The second EOF mode (Fig. R2c) is characterized by a circumglobal wave pattern and an equatorward shift of the westerly jet in the midlatitudes, exhibiting some resemblance to the teleconnection pattern induced by increased Asian AAs (Fig. 2b in the revision). The corresponding PC2 time series (Fig. R2d) also exhibits a similar temporal evolution as Asian AAs (Fig. R1a): an increase from 1990 to 2010 and then a decrease afterward. Regarding circulation changes over the subpolar North Atlantic, the leading EOF of SLP (Fig. R3a), resembling the negative phase of NAO, explains 50.9% of the total variance. The corresponding PC1 (Fig. R3b) also mirrors the evolution seen in Asian AAs since the 1990s (Fig. R1a), indicating weakened (strengthened) westerly winds over the Labrador Sea from 1990-2010 (2010-present).

Despite the above consistency in teleconnection patterns between reanalysis data and the responses induced by Asian AAs, *caution* is warranted in establishing causal relationships for several reasons:

- i) Statistical analysis alone cannot determine the causality, as the relationship can be influenced by common drivers or indirect links via a third process, which is common in climate science (e.g., Runge et al., 2019).
- ii) The sparse observational record of AAs (since the 1980s) and the AMOC (since the 2000s for RAPID, and 2014-2020 for OSNAP) makes it difficult to discern changes and attributions over a relatively short period.
- iii) Year-to-year changes in the mid-latitude atmospheric circulation are strongly affected by internal variabilities such as ENSO, North Atlantic Oscillation (NAO), and North Pacific Oscillation (NPO). The atmospheric conditions in the subpolar North Atlantic, the focus of this study, are strongly modulated by the NAO (e.g., Visbeck et al. 2001). Therefore, on interannual timescales, the contribution of forcing by AAs or even GHGs is not comparable to these internal variabilities. Even on decadal to interdecadal timescales, both the PC2 of the geopotential height (Fig. R2d) and the PC1 of the SLP (Fig. R3b) exhibit notable interdecadal variabilities, particularly in correlation with the Atlantic Multidecadal Oscillation (positive/negative phase after/before the 1990s; $r=0.60$ and 0.76 , respectively). This correlation suggests the possibility that the observed circumglobal wave pattern and the NAO-like response could be manifestations of internal variability.

To reduce the impact of internal variability and to establish causality, an ideal approach is to use large ensembles of simulations, each initialized with different conditions but subjected to the same AAs forcing over Asia. This methodology allows for a more accurate assessment of the individual impacts of Asian AAs and GHGs, and has been adopted in DAMIP and numerous other studies to support climate change detection and attribution assessments (e.g., Dong et al. 2021, 2022; Menary et al. 2020). Following this methodology, we have collected multiple lines of supporting evidence from CMIP6 and CESM2-LE simulations to demonstrate that the proposed link is not unique to CESM1, as detailed in our response to your subsequent comment.

Figure R1 | a, Temporal evolution of AOD at 550 nm (unit: 1) averaged over Asia based on MERRA2 dataset. Note that the values for the years 1982, 1983, 1991, and 1992 have been replaced with the means of adjacent years in order to eliminate the global impact of volcanic eruptions of El Chichon and Mount Pinatubo. **b**, Regression of 200 hPa geopotential height onto the Asian AOD time series (unit: m/1). **c**, Regression of sea level pressure (SLP) onto the Asian AOD time series (unit: Pa/1). **d-f**, Same as a-c, but associated with global-mean GHGs concentration based on EDGAR v6.0 Greenhouse Gas Emissions dataset. **Regression analyses applied to time series of Asian AAs and GHGs show similar results.**

Figure R2 | a,c The first two EOFs of 200-hPa geopotential height based on MERRA2 dataset. **b,d**, the corresponding PC time series. **The 2nd EOF mode is characterized by a circumglobal wave pattern, and its corresponding PC2 shows similarities with the evolution of Asian AAs since the 1990s.**

Figure R3 | a,c The first two EOFs of SLP over the subpolar North Atlantic based on MERRA2 dataset. b,d, the corresponding PC time series. **The 1st EOF mode is characterized by a negative NAO-like pattern, and its corresponding PC1 shows similarities with the evolution of Asian AAs since the 1990s.**

References

- Dong, Buwen, and Rowan T. Sutton. 2021. “Recent Trends in Summer Atmospheric Circulation in the North Atlantic/European Region: Is There a Role for Anthropogenic Aerosols?” *Journal of Climate* 34 (16): 6777–95. <https://doi.org/10.1175/JCLI-D-20-0665.1>.
- Dong, Buwen, Rowan T. Sutton, Len Shaffrey, and Ben Harvey. 2022. “Recent Decadal Weakening of the Summer Eurasian Westerly Jet Attributable to Anthropogenic Aerosol Emissions.” *Nature Communications* 13 (1): 1–10. <https://doi.org/10.1038/s41467-022-28816-5>.
- Fu, Qiang, and Pu Lin. 2011. “Poleward Shift of Subtropical Jets Inferred from Satellite-Observed Lower-Stratospheric Temperatures.” *Journal of Climate* 24 (21): 5597–5603. <https://doi.org/10.1175/JCLI-D-11-00027.1>.
- Lu, Jian, Gang Chen, and Dargan M. W. Frierson. 2010. “The Position of the Midlatitude Storm Track and Eddy-Driven Westerlies in Aquaplanet AGCMs.” *Journal of the Atmospheric Sciences* 67 (12): 3984–4000. <https://doi.org/10.1175/2010JAS3477.1>.
- Menary, Matthew B., Jon Robson, Richard P. Allan, Ben B.B. Booth, Christophe Cassou, Guillaume Gastineau, Jonathan Gregory, et al. 2020. “Aerosol-Forced AMOC Changes in CMIP6 Historical Simulations.” *Geophysical Research Letters* 47 (14). <https://doi.org/10.1029/2020GL088166>.
- Runge, J., Bathiany, S., Bollt, E., Camps-Valls, G., Coumou, D., Deyle, E., et al. (2019). Inferring causation from time series in Earth system sciences. *Nature Communications*, 10, 2553. <https://doi.org/10.1038/s41467-019-10105-3>

Visbeck, Martin H., James W. Hurrell, Lorenzo Polvani, and Heidi M. Cullen. 2001. “The North Atlantic Oscillation: Past, Present, and Future.” *Proceedings of the National Academy of Sciences of the United States of America* 98 (23): 12876–77. <https://doi.org/10.1073/pnas.231391598>.

3. It would be also useful and highly confirmatory to show if CMIP6 models altogether show a same connection of AA emissions changes over Asia and AMOC. Among the CMIP6 models, when there are different future AA emission levels or aerosol radiative effect (under the same scenario) over East Asia, do they show any correlation with their respective AMOC changes? The goal is if the authors could show somehow the model independency of the main findings and a similar connection from AA emissions over Asia to AMOC across multiple models.

Response: Thank you for your valuable suggestion. In the revision, we have incorporated new analyses and discussions to provide additional multi-model validation of the Asian AAs-AMOC link:

i) Validation of teleconnection patterns through AerChemMIP: To assess the robustness of the atmospheric teleconnections induced by increased Asian AAs, we have analyzed simulations from 10 models participating in the Aerosol Chemistry Model Intercomparison Project (AerChemMIP) (Collins et al. 2017), a subset of CMIP6. Two sets of AerChemMIP simulations, piClim-control and piClim-SO₂, were used. Both simulations span 30 years with fixed climatological mean SST and sea ice conditions corresponding to the pre-industrial levels. The only difference is in the emissions of anthropogenic aerosol precursors of SO₂: piClim-control uses preindustrial emissions, while piClim-SO₂ employs present-day (2014) SO₂ emissions. Notably, due to the implementation of the Clean Air Act in Europe and North America, SO₂ emissions in the western hemisphere had already significantly decreased by 2014, making Asian emissions the dominant forcing in the piClim-SO₂ simulations. Therefore, this 2014 SO₂ forcing (Fig. R4a, contours), albeit being more concentrated in East Asia, operates as a comparable agent to that used in our EAST simulations.

Our analysis reveals consistent patterns of atmospheric response in the AerChemMIP simulations (Fig. R4) compared to our EAST simulations (Fig. 2). These patterns include a circumglobal teleconnection pattern (Fig. R4a) and an equatorward migration of the westerly jet in the northern midlatitudes (Fig. R4c). Additionally, a negative NAO-like response is evident in the subpolar North Atlantic, accompanied by suppressed westerlies over the Labrador Sea (Fig. R4b). This robust agreement between the results of AerChemMIP and our CESM1 simulations provides confidence in the robustness of the atmospheric teleconnections excited by increased Asian AAs. However, due to the absence of ocean models in the AerChemMIP simulations, further examination of AMOC responses is not feasible with these simulations.

ii) Validation of AMOC responses through CESM2-LE: Recognizing the limitations of the AerChemMIP simulations in examining AMOC responses and the challenges of disentangling aerosol changes in the eastern and western hemispheres in DAMIP experiments, we turn to the CESM2 Large Ensemble (CESM2-LE) (Simpson et al. 2023). This ensemble extends its AAs-alone ensemble simulations with 15 members into the year 2050 under the shared socioeconomic pathways 3-7.0 (SSP3-7.0) scenario.

In this high-emission scenario, Asian AA emissions continue to rise in the future, while AAs in the western hemisphere show limited changes as they have already diminished to low levels. Consequently, the increased Asian AAs and the associated radiative cooling emerge as the dominant forcing during the period 2015 to 2050 (Fig. R5b, contours). This provides analogs to our EAST results and adds some model diversity to the analysis. The circumglobal teleconnection pattern (Fig. R5b, color) and the negative NAO-like response (Fig. R5c) are also found in the CESM2-LE simulations. In addition, the CESM2-LE simulations show that the weakened westerly over the Labrador Sea suppresses the heat loss, resulting in a rapid decline of AMOC during the period 2015 to 2035 (Fig. R5a). Such agreement across different models suggests that the relationship between Asian AAs and AMOC is not unique to CESM1, and supports our argument that the increased Asian AAs can weaken the AMOC by exciting atmospheric teleconnections and hindering deep water formation at the deep convection site.

In the revision, the above analysis and related discussion have been added as a new section entitled "**Additional multi-model results confirming the Asian AAs-AMOC link**" before the conclusion (Lines 237-292). We are delighted at the consistency between different models that provides a compelling line of evidence for our Asian AAs-AMOC link. Thank you for the suggestions.

Figure R4 | Responses in atmospheric circulation to 2014 sulfate emission forcing in AerChemMIP simulations. **a**, Responses of aerosol optical depth (contours; unit: 1), 200-hPa streamfunction (colors; unit: $10^6 m^2 s^{-1}$) and rotational wind (vectors; unit: $m s^{-1}$) calculated as the difference between the piClim-SO2 ensemble and the piClim-control ensemble for years 1-30. **b**, Responses of sea level pressure (colors; unit: Pa) and surface wind (vector; unit: $m s^{-1}$) over the North Atlantic of the multi-model mean. Blue and black vectors denote anomalous winds that weaken and intensify the climatological winds. **c**, Response of the zonal-mean zonal winds (colors; unit: $m s^{-1}$) of the multi-model mean, and superimposed is the corresponding climatology in piClim-control. **The Asian AAs-induced atmospheric responses are validated by the AerChemMIP simulations.**

Figure R5 | Responses in atmospheric circulation and AMOC to increasing Asian AAs in CESM2-SF-LE simulations. **a**, AMOC time series (unit: Sv) in CESM2-SF-LE SSP370-AAs simulations. The AMOC index is obtained as the maximum of streamfunction between latitudes 25° N–75° N and between depths of 500 and 2000 m. **b–d**, Same as Supplementary Fig. 9a–c, but for the difference between years 2040–2050 and years 2015–2025 of the ensemble mean from SSP370-AAs simulations. **The Asian AAs-AMOC link is validated by the CESM2-SF-LE simulations.**

References

- Collins, J. William, Jean François Lamarque, Michael Schulz, Olivier Boucher, Veronika Eyring, I. Michaela Hegglin, Amanda Maycock, et al. 2017. “AerChemMIP: Quantifying the Effects of Chemistry and Aerosols in CMIP6.” *Geoscientific Model Development* 10 (2): 585–607. <https://doi.org/10.5194/gmd-10-585-2017>.
- Simpson, Isla R., Nan Rosenbloom, Gokhan Danabasoglu, Clara Deser, Stephen G. Yeager, Christina S. McCluskey, Ryohei Yamaguchi, et al. 2023. “The CESM2 Single-Forcing Large Ensemble and Comparison to CESM1: Implications for Experimental Design.” *Journal of Climate* 36 (17): 5687–5711. <https://doi.org/10.1175/JCLI-D-22-0666.1>.

Specific comments:

Line 76: what do you mean by “present day level”?

Response: We apologize for not explaining this more clearly in the manuscript. “present-day level” refers to the levels of GHGs, aerosols, and solar insolation observed in the year 2000. For example, the global mean GHG mixing ratio is 367.0 ppmv. This clarification has been included in the revised text (Line 83).

Line 77: why 10% ?

Response: The choice of a 10% perturbation was made to generate a robust response signal in our simulations. This level of perturbation results in a radiative forcing amplitude of approximately 30 W/m², which is confined to specific regions over Asia, North America, and Western Europe. The rationale for using a stronger forcing is to ensure that the effects of Asian aerosols on the AMOC can be clearly distinguished from the background noise introduced by natural variability, such as the Atlantic Multidecadal Variability (AMV). We have incorporated this explanation into the revised manuscript (Lines 88-90).

Line 80: here the response is reversed, which means the authors assume a linear nature of the response. This has to be stated and if the authors could provide some evidence behind such an assumption that would be even better.

Response: Thank you for your comment, and we have clarified that a linear assumption is adopted in the revision (Line 86). Your point has inspired us to consider the linearity of the AMOC response to changes in AAs, which we plan to explore in future research.

Line 85: why only 10 years of simulations are performed? This is quite small and unusual. I understand that the response time for aerosol would be small but a time of emergence analysis on the AMOC strength should be done considering its variability in the model to determine the minimum duration of the run to find a discernible forced signal from internal variability. I am okay with the claims made qualitatively from this limited length data, though the quantification could be different and could show a further robust estimate if the runs are performed for longer times. I would urge the authors to do this if there is no major computational constraint or put some strong reasoning to perform such short simulations. It is not ideal to set such experimental precedence without any strong reasoning.

Response: We agree with your point about the length of the simulation. In the original manuscript, we presented the ensemble simulations for the *first 20 years* because our primary focus was on how Asian radiative cooling affects atmospheric circulation and leads to a rapid reduction of the AMOC during the first two decades. Beyond this period, the AMOC tends to weaken at a considerably slower rate. This deceleration in AMOC weakening is attributed to the anomalous cold advection associated with a weakened AMOC, which acts to balance the surface heat gain occurring in the Labrador Sea and suppress further decrease of the deep water formation (please see details in our response to your next comment).

Taking your suggestion, we have extended the simulation period to 100 years for all ensemble members. In Fig. R6, we present the updated temporal evolution of the AMOC index in the EAST and WEST experiments, and the main findings remain consistent. In the EAST experiments, the AMOC experiences a rapid decline of ~ 2.7 Sv in the first four decades, followed by a slower decline of ~ 0.6 Sv over the subsequent 60 years. In the WEST experiments, the AMOC is weakened by ~ 1.4 Sv in the first four decades and remains relatively stable at this level for the next 60 years. Importantly, the eastern hemisphere forcing still exhibits a greater efficacy in weakening the AMOC compared to the adjacent western hemisphere forcing. Furthermore, the spatial patterns of responses of the AMOC and atmospheric circulations averaged over the last 50 years (Fig. R7) are consistent with those averaged over years 1-40 (Figs. 2b-c,3b), again demonstrating a short response timescale of the AMOC to aerosol changes.

In the revision, we still focus on the transient responses to emphasize how increased Asian AAs weaken the AMOC through the excitation of atmospheric teleconnections. However, the results are updated based on the average over the first 40 years. In addition, we provide an overview of the *ensemble* responses during the slower response stage (years 51-100) in the supplementary, demonstrating the limited changes in response patterns over time. We have revised the figures and the corresponding text in the revision (Lines 107-113).

Fig. R6 a,b Ensemble-mean evolution of annual-mean AMOC index (unit: Sv) in EAST and WEST smoothed with a 5-yr running-mean filter, with the thin curves denoting each member's anomalies. **The radiative forcing due to changes in aerosols over both Asia and North America & Europe causes the AMOC to slowdown.**

Fig. R7 Responses of atmospheric and deep water formation to increased Asian AAs during the slow response stage. a, Responses of 200-hPa streamfunction (colors; unit: $10^6 \text{ m}^2 \text{ s}^{-1}$) and rotational wind (vectors; unit: m s^{-1}) in the northern hemisphere averaged over year 51-100. **b,** Responses of sea level pressure (colors; unit: Pa) and surface wind (vector; unit: m s^{-1}) over the North Atlantic in EAST simulations averaged over year 51-100. **c,** WMT responses (unit: Sv) over the subpolar North Atlantic (SPNA) and contributions from Irminger Sea, Labrador Sea, and Greenland-Iceland-Norwegian Seas (GIN). **The long-term responses of atmospheric circulation and water mass transformation show consistent patterns compared to the transient responses.**

Line 98: Why is there a rapid decrease in the first two decades? Could the authors give a reason/hypothesis for this?

Response: The AMOC response, as depicted in Fig. R6a, exhibits two distinct timescales: a transient phase during years 1-40 and a slow phase from years 41-100. During the transient phase of the response, the atmosphere acts primarily as a forcing on the ocean, and the feedback from ocean dynamical processes is relatively minor, and the weakened westerlies lead to reduced heat loss from the surface of the Labrador Sea, effectively weakening the AMOC strength.

However, the importance of the ocean's feedback is increasing as time progresses into the slow response phase. Since the AMOC is responsible for the majority of the mean heat transport in the

Atlantic basin, its weakening results in reduced oceanic poleward heat transport (Fig. R8a). Consequently, anomalous colder water (Fig. R8b) and positive density anomalies propagate downstream along the cyclonic subpolar gyre, reaching the Labrador Sea. This process balances the surface heat gain in the Labrador Sea, preventing further reduction of deep water formation and thus alleviating the weakening of the AMOC. This above explanation has been included in the revised manuscript to provide a clearer understanding of the dynamics involved (Lines 228-235).

Fig. R8 a. Ensemble-mean evolution of oceanic northward heat transport at 40N (unit: PW) in EAST smoothed with a 5-yr running-mean filter, with the thin curves denoting each member’s anomalies. **b.** SST response (unit: °C) in EAST averaged over year 51-100. **The weakened AMOC reduces the northward oceanic heat transport and advects anomalously cold water into the deep convection site, balancing the surface heat gain there.**

Line 100: Similarly, why is there a gradual decline afterwards?

Response: As mentioned in our previous response, the gradual decline of the AMOC during the slow response phase is a result of the atmospheric forcing and oceanic feedback processes, with the latter contributing to a decreased northward heat transport.

Line 108-111: Are these separate experiments performed with only South Asian AA and East Asian AA? The description of such experiments should be clearly stated then and the supplementary figure 3 also do not provide the images of two separate experiments. Hence it remains unclear what the authors have done.

Response: We apologize for not explaining this clearly in the manuscript. In addition to the EAST experiment that represents the total impact of Asian AAs, in this study, we also performed the East Asia experiment and inferred the influence of South Asian AAs by comparing it to the EAST experiment. The relative contribution of these two regions, East Asia and South Asia, to AMOC changes is an interesting and complex issue, as East Asia has experienced a decrease in AAs emissions since the 2010s, while South Asia has experienced a continuous increase in AAs emissions.

However, we are aware that a single member of the simulation is insufficient to accurately estimate the contributions of either East Asia or South Asia. Given the scope of this study, we have decided to focus on the impact of increased Asian AAs as a whole on AMOC strength and leave for the

future a more in-depth investigation of the relative role of East Asia and South Asia on AMOC strength changes.

Line 145: please remove “greatly”. And there are key differences in the CGT pattern and the CGT-like pattern that the authors find in this experiment. That has been pointed out too. I am curious if the authors could show such resembling pattern in observations with respect to observed Asian AA variations.

Response: Thank you for pointing this out. We have removed "greatly" from the sentence as suggested and simplified the terminology to "circumglobal response" instead of “CGT-like response” to describe the wave pattern identified in the experiments.

In addition, as shown in Fig. R2c, the second EOF of observed geopotential height is characterized by a circumglobal wave pattern, exhibiting a resemblance to the teleconnection pattern induced by increased Asian AAs. The corresponding PC2 time series (Fig. R2d) also exhibits some similarity to the temporal evolution of Asian AAs since the 1990s (Fig. R1a). However, as mentioned in our response to your major comment #2, the regression analysis cannot establish causality. Besides, the observed circumglobal wave pattern could be a manifestation of internal variability.

Line 150: is there an associated seasonal dependency in the AMOC response to Asian AA?

Response: Not much, and the AMOC weakening is evident in all seasons (Fig. R9), likely due to the significantly longer response timescales associated with the deep western boundary currents and horizontal recirculations that collectively constitute the AMOC (Zhang 2010).

Fig. R9 a-d, Seasonal mean responses of AMOC to increased Asian AAs. **e**, Seasonal mean streamfunction anomaly averaged over 50-65N, 36.75-37.0 kg m⁻³. **The AMOC weakening is evident in all four seasons.**

Reference

Zhang, Rong. 2010. “Latitudinal Dependence of Atlantic Meridional Overturning Circulation (AMOC) Variations.” *Geophysical Research Letters* 37 (16): 1–6. <https://doi.org/10.1029/2010GL044474>.

Figure 2c : in this figure the location of the mean westerlies in contours must be shown.

Response: Thank you for your comment, and the mean zonal wind at 200 hpa has been superimposed in the revised Fig. 3b to highlight the location of the waveguide.

Line 165-167: here the authors claim that previous studies from CMIP5 and CMIP6 models already corroborate the link of the equatorward shift of the jet and AA emission over Asia. In a similar line, could the authors also link the changes in AA over Asia and AMOC changes in those CMIP5 and CMIP6 models?

Response: We apologize for the incorrect description in our original manuscript. To clarify, previous studies based on observations and CMIP models have indeed reported the equatorward shift of Atlantic jets from the 1970s to the 2010s (Dong et al. 2021). However, the link between this shift and AAs changes has only been established through DAMIP single forcing experiments (Undorf et al. 2018; Dong et al. 2022) or purposefully designed atmospheric experiments (Dong et al. 2021). In other words, these studies did not explicitly attribute these changes to AAs over Asia, as AA emissions from North America and Europe exhibit significant variations during the same time period. On the other hand, using the CESM1, Diao et al. (2021) attributed a central role to Asian AAs in driving the equatorward shift of the jet, but the atmospheric changes in the subpolar North Atlantic didn't receive attention in their study, let alone the impact on the AMOC.

We have revised the text accordingly in the manuscript (Lines 168-170) to provide a more accurate representation of the previous studies and their findings. In addition, as in our response to your major comment #3, we have made an extensive effort to explore the atmospheric circulation and AMOC response to AAs changes in AerChemMIP simulations and CESM2-SF-LE simulations. All of these modeling results provide robust corroborating evidence for our findings. Please see our new section in the revision “**Additional multi-model evidence confirming the Asian AAs-AMOC link**” for details.

References

- Diao, Chenrui, Yangyang Xu, and Shang Ping Xie. 2021. “Anthropogenic Aerosol Effects on Tropospheric Circulation and Sea Surface Temperature (1980-2020): Separating the Role of Zonally Asymmetric Forcings.” *Atmospheric Chemistry and Physics* 21 (24): 18499–518. <https://doi.org/10.5194/acp-21-18499-2021>.
- Dong, Buwen, and Rowan T. Sutton. 2021. “Recent Trends in Summer Atmospheric Circulation in the North Atlantic/European Region: Is There a Role for Anthropogenic Aerosols?” *Journal of Climate* 34 (16): 6777–95. <https://doi.org/10.1175/JCLI-D-20-0665.1>.

Dong, Buwen, Rowan T. Sutton, Len Shaffrey, and Ben Harvey. 2022. “Recent Decadal Weakening of the Summer Eurasian Westerly Jet Attributable to Anthropogenic Aerosol Emissions.” *Nature Communications* 13 (1): 1–10. <https://doi.org/10.1038/s41467-022-28816-5>.

Undorf, Sabine, M. A. Bollasina, and G. C. Hegerl. 2018. “Impacts of the 1900-74 Increase in Anthropogenic Aerosol Emissions from North America and Europe on Eurasian Summer Climate.” *Journal of Climate* 31 (20): 8381–99. <https://doi.org/10.1175/JCLI-D-17-0850.1>.

Line 180: these are model specific behaviour of the AMOC and authors must be careful to generalise such things. Could the author confirm the same in observations? Lozier et.al. suggest that observed AMOC behaves in a different fashion than what the models show and AMOC is more sensitive to the changes in the eastern side of subpolar North Atlantic than the western side over Labrador Sea. Hence this potential difference in observed and model behaviour and its effect on the results needs to be clarified.

Response: Thank you for your comment on this. As addressed in our response to your major comment #2, the deep convection in the LS is overestimated in climate models, but it has been found in recent studies (Yeager et al. 2021; Oldenburg et al. 2021, 2022) to drive AMOC low-frequency variability, even in high-resolution models that exhibit reasonable consistency with the weak overturning in the OSNAP observations.

We have added a discussion in the revised manuscript (Lines 186-188) that highlights the debate over the role of the Labrador Sea in the AMOC and the potential differences between modeled and observed behavior. We also cite relevant studies, including Lozier et al. (2019), to acknowledge the variability in AMOC sensitivity in the revision.

Line 214: could the authors show the changes in horizontal (zonal and meridional) temperature advection to confirm that the less cold air is getting advected over the Labrador Sea from North America? This analysis would complete the picture by clearly showing anomalous horizontal warm air advection.

Response: Thanks for the comments. Following your suggestion, we have decomposed the horizontal advection terms into contributions from zonal and meridional processes in Supplementary Fig. 7e-f, and it does indeed illustrate the changes in zonal temperature advection dominates, highlighting the less cold air getting advected over the Labrador Sea from North America. We have discussed this in the revised manuscript (Line 220).

Line 470: cures  curves

Response: Thank you, and it has been corrected.

Reviewer #2 (Remarks to the Author):

Key Results

Recent work has highlighted how aerosols over the North Atlantic region can strengthen AMOC by cooling the deep water formation region, which strengthens meridional density gradients driving AMOC. This study finds a new and potentially important contrary effect: aerosols over South Asia and East Asia can weaken AMOC by shifting the Atlantic Westerly winds southward, decreasing heat loss and warming the deep water formation regions. Heat budget analysis makes it clear that the main mechanism is the reduction in transport of cold continental air over the Labrador Sea, which reduces ocean heat loss, cooling, and density increase there.

Validity, Data and Methodology

The authors themselves point out what I see as the main question mark about the validity of the result, which is that climate models such as theirs do not do a good job capturing the overflow component (from the Nordic Seas) of deep water formation and hence may over-emphasize the role of the Labrador Sea. Since this is a very challenging feature to model correctly, it is a shortcoming of virtually all studies of AMOC, and should not hinder publication.

Response: We appreciate your understanding of this challenge, which is indeed a common limitation in AMOC studies due to the complex nature of modeling deep water formation processes in the North Atlantic.

It's worth noting that while the climate model may not simulate the mean overturning well, recent studies (Yeager et al. 2021; Oldenburg et al. 2021, 2022) have highlighted the role of the Labrador Sea in driving long-term changes in deep convection and AMOC, even in high-resolution models that exhibit reasonable consistency with the weak mean overturning in the OSNAP observations. In light of this, we believe that our study provides valuable insights into how increased Asian AAs may contribute to the weakening of the AMOC. Thank you for recognizing the importance of our study despite this acknowledged limitation.

References

- Oldenburg, Dylan, Robert C.J. Wills, Kyle C. Armour, Luanne Thompson, and Laura C. Jackson. 2021. "Mechanisms of Low-Frequency Variability in North Atlantic Ocean Heat Transport and AMOC." *Journal of Climate* 34 (12): 4733–55. <https://doi.org/10.1175/JCLI-D-20-0614.1>.
- Oldenburg, Dylan, Robert C.J. Wills, Kyle C. Armour, and Lu Anne Thompson. 2022. "Resolution Dependence of Atmosphere–Ocean Interactions and Water Mass Transformation in the North Atlantic." *Journal of Geophysical Research: Oceans* 127 (4): 1–18. <https://doi.org/10.1029/2021JC018102>.
- Yeager, Stephen, Fred Castruccio, Ping Chang, Gokhan Danabasoglu, Elizabeth Maroon, Justin Small, Hong Wang, Lixin Wu, and Shaoqing Zhang. 2021. "An Outsized Role for the

Labrador Sea in the Multidecadal Variability of the Atlantic Overturning Circulation.”
Science Advances 7 (41). <https://doi.org/10.1126/sciadv.abh3592>.

Significance

In recent decades, the largest aerosol increases have been in Asia, and this study is the first to show a clear link between these increases and possible weakening of AMOC. Therefore it is a highly important study. The fact that there has been increasing discussion of weakening of AMOC, presumably due to the effects of anthropogenic global heating, will heighten interest in this paper because it suggests an alternative explanation for dropping AMOC strength.

Response: We sincerely appreciate your recognition.

Clarity & Context/ Suggested Improvements

The paper is very well written. I have some comments on what is written about the context which also has some bearing on how the paper presents the significance of these findings.

1) Previous work on Asian Aerosols and teleconnections to N Atl

The last paragraph before the Discussion section discusses prior work, and it (or at least the first part of it, before the discussion of discrepancy with Kang et al and new results) should be placed at the end of the Introduction section. I assume the motivation for putting it later was that the paragraph is motivated by this paper’s results that the main influence on AMOC of the Asian AA is through a shift of the westerlies. However, the paragraph (if placed in Intro) would make sense to a reader since it talks about effects of climate perturbations in Asia on the North Atlantic region, which is plausibly relevant to the topic at hand.

One sentence in that paragraph makes me think that this paper should summarize its results a little differently. The sentence says “experiments forced with ... aerosol emissions over the eastern hemisphere... has shown a weakening of the AMOC... providing direct support for our finding.” If that were true, doesn’t the existence an earlier paper with similar results call into question the novelty of the finding? I have not thoroughly read the paper in question (Diao et al, 2021), but as far as I can tell, Diao et al find the Asian aerosol effect on moving the Westerlies southward over the Atlantic, but not the further effect on the AMOC.

Therefore, the abstract and Intro should be rephrased a bit to make it clear that this study finds that a previously-discovered shift in Westerlies makes the Asian AA have the effect of decreasing AMOC.

Response: Thank you for your suggestions. We agree with you to move the discussion of Diao et al. (2021) to the Introduction. We have made the necessary revisions to both the Abstract and Introduction to better introduce the topic and provide context for the readers (Lines 66-72).

Regarding our findings relative to previous work, we appreciate the opportunity to clarify. While Diao et al. (2021) indeed identified the role of increased Asian aerosols in shifting the westerlies, they did not explore the further impact on the AMOC. To validate the results of our experiments with TOA radiative forcings, we requested the AMOC output from the authors of Diao et al. (2021) and examined the AMOC response, as shown in the original supplementary Fig. 8. To avoid any potential confusion, in the revised manuscript we have refrained from presenting the AMOC response derived from the data of Diao et al. (2021).

Given that both Diao et al. (2021) and our study employ the same model CESM1, a better way to validate our findings is through multi-model or large-ensemble simulations. In the revision, we comprehensively examined the atmospheric and AMOC responses to increased Asian AAs using the AerChemMIP and CESM2 single-forcing large ensembles. These independent analyses yielded results consistent with our study, including a circumglobal teleconnection pattern, a negative NAO-like response in the North Atlantic, reduced heat loss in the Labrador Sea, and a weakening of the AMOC. In the revised manuscript, we have added a new section entitled "**Additional multi-model results confirming the Asian AAs-AMOC link**" before the conclusion section (Lines 237-292) to emphasize the consistency between different models in supporting the proposed link between Asian aerosols and AMOC changes.

2) Are the new results surprising?

This is more subjective, but I don't think the result is quite as surprising as they portray. The strength of the AMOC depends less on density than on density differences between the northern North Atlantic and other parts of the Ocean (see Klinger and Haine, Ocean Circulation in Three Dimensions, Chapter 9). Thus, the fact that cooling the North Atlantic with aerosols strengthens AMOC does not imply that cooling a different latitude and longitude with aerosols should have the same affect. The Kang et al (2021) study cited here does suggest opposing AMOC increase rather than decrease, so maybe the real question this study is raising is not "what does Asian AA forcing do to AMOC?" but "what does Asian AA forcing at lower latitudes do to AMOC?"

Response: We agree with you that radiative cooling at different latitudes can lead to different climatic impacts. The mechanism through which the Asian AAs forcing impacts the AMOC in Kang et al. (2021) relies on downstream advective processes, in contrast to our study where we emphasize the importance of exciting Rossby waves in facilitating this influence. The specific location of the increased Asian AAs along the path of the mean subtropical jet is a crucial factor that enhances their influence on altering atmospheric circulation patterns and, by extension, the AMOC. For radiative forcings positioned at the same latitude but outside of the waveguide, their impact on the AMOC could be very different.

In the revision, we have moved the comparison with Kang et al. (2021) to the last section (Lines 302-308) and highlighted the sensitivity of the global climate system to the location of the forcing.

3) Some minor suggestions

I recommend changing name of last section to "Discussion and Conclusions" and moving the comparison to earlier studies (end of last paragraph before discussion) to this section.

Response: Thank you. Following your suggestions, we have updated the name of the last section to "Discussion and Conclusions" and moved the comparison with previous studies to this section (Lines 302-308).

REVIEWERS' COMMENTS

Reviewer #1 (Remarks to the Author):

The authors have thoroughly revised the article based on my previous comments. I am more than satisfied by the arguments provided by the authors and totally agree with the limitations mentioned. The results are promising and unique and would surely make an impact on the ongoing research in this field. I recommend acceptance of this manuscript and congratulate for this fine work.

Reviewer #2 (Remarks to the Author):

For original submission, I believe I recommended publication subject to response to my comments. I am satisfied with the response and now recommend publication.